# Self-Sustained Collective Motion of Two Joint Liquid Crystal Elastomer Spring Oscillator Powered by Steady Illumination

**DOI:** 10.3390/mi13020271

**Published:** 2022-02-08

**Authors:** Changshen Du, Quanbao Cheng, Kai Li, Yong Yu

**Affiliations:** Department of Civil Engineering, Anhui Jianzhu University, Hefei 230601, China; changshendu@yeah.net (C.D.); cheng_quanbao@outlook.com (Q.C.); kli@ahjzu.edu.cn (K.L.)

**Keywords:** spring oscillator, liquid crystal elastomer, collective motion, domain of attraction

## Abstract

For complex micro-active machines or micro-robotics, it is crucial to clarify the coupling and collective motion of their multiple self-oscillators. In this article, we construct two joint liquid crystal elastomer (LCE) spring oscillators connected by a spring and theoretically investigate their collective motion based on a well-established dynamic LCE model. The numerical calculations show that the coupled system has three steady synchronization modes: in-phase mode, anti-phase mode, and non-phase-locked mode, and the in-phase mode is more easily achieved than the anti-phase mode and the non-phase-locked mode. Meanwhile, the self-excited oscillation mechanism is elucidated by the competition between network that is achieved by the driving force and the damping dissipation. Furthermore, the phase diagram of three steady synchronization modes under different coupling stiffness and different initial states is given. The effects of several key physical quantities on the amplitude and frequency of the three synchronization modes are studied in detail, and the equivalent systems of in-phase mode and anti-phase mode are proposed. The study of the coupled LCE spring oscillators will deepen people’s understanding of collective motion and has potential applications in the fields of micro-active machines and micro-robots with multiple coupled self-oscillators.

## 1. Introduction

Self-excited oscillation is a kind of periodic motion that is maintained by constant external excitations [1,2,3,4]. Similar to biological active feeding, it can directly harvest energy from a constant environment to maintain its periodic motion [5,6]. In addition, the period and amplitude of the self-oscillation generally depend on the intrinsic parameters of the system and are independent of the initial conditions, which makes the system robust [7,8,9]. Due to the unique advantages of self-excited oscillation systems, they have broad application prospects in the fields of energy acquisition, sensing with electronic skins [10], soft robotics [11,12,13], medical instruments [14,15,16,17], and motors [18]. The self-excited oscillations are generally based on responsive materials, including liquid crystal elastomers (LCEs) [19,20], dielectric elastomers [21], hydrogels [22,23,24], and ion gels [25,26]. Based on different stimuli-responsive materials and structures, different feedback mechanisms are proposed to realize energy compensation, such as the coupling of chemical reactions and large deformation [25,26], the self-shadowing effect [14,27], the coupling of liquid volatilization, and membrane deformation [28].

Based on the self-oscillation systems that have been previously reported [29], the coupling and synchronization phenomena of two or more self-excited oscillation systems and their collective motion have attracted extensive attention [30,31,32,33,34]. Synchronization and collective motion are ubiquitous in nature, such as the overall movement of a school of fish or a flock of wild geese or a group of fireflies flickering together [35,36,37]. As early as 1673, C. Huygens studied the synchronization phenomenon of simple pendulum coupling by observing two identical clocks oscillating synchronously with two pendulums swinging in opposite directions [38]. Recent research has confirmed that the coupling between two pendulums is caused by tiny mechanical oscillations that propagate through the wooden structure in which the clocks are mounted. Similar experiments have enabled a large number of metronomes to swing synchronously on a freely moving base [39,40]. Recently, based on optically responsive LCE, Ghislaine et al. experimentally studied the synchronized oscillations of thin plastic actuators fueled by light and found two kinds of in-phase and anti-phase synchronous oscillation phenomena in the steady-state [30,31]. Their numerical simulations qualitatively explained the origin of synchronized motion and found that motion can be regulated by the mechanical properties of coupling.

LCEs are advanced multifunctional materials that combine the flexibility of polymeric networks with the nematic structure of liquid crystals [41,42], which have the advantages of having a fast response, recoverable deformation, and low noise [10,43,44,45]. This special composition and structure enable LCEs to respond to external light [46,47,48,49], heat [50], electric fields [51,52], magnetic fields [53] and chemical substances [54]. Based on LCE materials, several self-exciting motion modes have been constructed, such as rolling [20], vibration [17], swinging [10,55], stretching and shrinking [56], rotation [57], eversion or inversion [9,58], torsion [59], jumping [60], and buckling [61] modes. These self-exciting motion modes provide good ideas for studying the coupling of multiple systems and their collective motion.

Studying the self-excited oscillation coupling of two or more systems and their collective motion is beneficial to the construction of richer and more complex types of motion, allowing them to demonstrate more versatile functions in the micro-robots. In view of this, we used two identical LCE fibers connected by a spring to construct a new optically responsive LCE spring oscillation coupling system. The self-oscillation mechanism and the possible in-phase mode, anti-phase mode, and non-phase-locked mode are discussed, and the influence of some key physical quantities on its synchronization modes, amplitude, and period are analyzed. The layout of this paper is as follows. In Section 2, based on the LCE dynamic model first proposed by Finkelmann et al. [62,63], the governing equation is derived, and the difference schemes of the dynamics equation and the solution method are given. In Section 3, the different synchronization modes of the self-excited oscillation are discussed, and the detailed mechanisms are revealed. In Section 4, the effects of various system parameters on the self-excited oscillation in the three synchronization modes are studied in detail. In Section 5, the equivalent systems in the in-phase mode and anti-phase mode are provided. Finally, the concluding remarks are shared in Section 6.

## 2. Model and Formulation

### 2.1. Dynamic Model of the Two LCE Spring Oscillators

Figure 1 sketches a dynamic model of the self-oscillation coupling system under uniform and constant illumination, which is composed of two identical LCE spring oscillators and a spring. The illuminated zone is represented by the shaded area. The original length of the LCE fiber in a stress-free state is $L_{f}$, and the original length of the spring is $L_{s}$, as shown in Figure 1a. Next, fix one end of the LCE fiber and tie the mass on the other end, and connect the two fibers with a spring. For simplicity, the gravity of the mass blocks is ignored, and an initial strain $\lambda_{p}-1$ is assigned to the LCE fibers to ensure that the system hang horizontally, as shown in Figure 1b. The displacement of the two mass blocks is denoted by $u_{1}\left(t\right)$ and $u_{2}\left(t\right)$, respectively. The current lengths of LCE fiber 1 and LCE fiber 2 are $l_{f1}\left(t\right)=L_{f}\lambda_{p}+u_{1}\left(t\right)$ and $l_{f2}\left(t\right)=L_{f}\lambda_{p}-u_{2}\left(t\right)$, respectively, as shown in Figure 1c. $F_{f1}\left(t\right)$ and $F_{f2}\left(t\right)$ are the spring forces of LCE fiber 1 and LCE fiber 2, respectively, which are also called driving forces hereinafter. $F_{s}\left(t\right)$ is the spring force of the spring (abbreviated as spring force), and $F_{d1}\left(t\right)$ and $F_{d2}\left(t\right)$ are the damping forces during oscillation. For simplicity, it is assumed that the damping forces are proportional to the velocity of the mass, and the direction is always opposite to the velocity of the mass.

In order to analyze the inhomogeneous deformation of the two LCE fibers, the Lagrangian coordinate systems $X_{1}$ and $X_{2}$ are fixed and established for the initial configuration of LCE fiber 1 and LCE fiber 2, and the Eulerian coordinate systems $x_{1}$ and $x_{2}$ in the current configuration are also established. The instantaneous position of a material point $X_{1}$ ($X_{2}$) of LCE fiber 1 (LCE fiber 2) can be represented as $x_{1}=x_{1}\left(X_{1},t\right)$ ($x_{2}=x_{2}\left(X_{2},t\right)$) during the oscillation. Since the pulling forces of the LCE fibers are much greater than the gravity of the mass blocks and the LCE fibers, and we ignore the gravity of the mass blocks and the LCE fibers for simplicity. According to Newtonian mechanics, the following governing equation holds at any moment during mass oscillation.
(1)$$\left\{\begin{array}{l} m{\ddot{u}}_{1}=-F_{f1}\left(t\right)+F_{s}\left(t\right)-c{\dot{u}}_{1}\\ m{\ddot{u}}_{2}=F_{f2}\left(t\right)-F_{s}\left(t\right)-c{\dot{u}}_{2} \end{array}\right.,$$
where c is the damping coefficient, $\dot{u}$ and $\ddot{u}$ indicate the velocity $\frac{du\left(t\right)}{dt}$ and acceleration $\frac{d^{2}u(t)}{dt^{2}}$ of the mass, respectively.

For simplicity, the force of the LCE fiber $F_{f}\left(t\right)$ is assumed to be proportional to the elastic strain $\varepsilon_{e}\left(t\right)$, i.e.,
(2)$$\left\{\begin{array}{l} F_{f1}\left(t\right)=K_{f}L_{f}\varepsilon_{e1}\left(t\right)\\ F_{f2}\left(t\right)=K_{f}L_{f}\varepsilon_{e2}\left(t\right) \end{array}\right.,\;$$
where $K_{f}$ is the spring constant of the LCE fiber, and $\varepsilon_{e}\left(t\right)=\varepsilon_{\mathrm{tot}}\left(t\right)-\varepsilon \left(t\right)$ with $\varepsilon_{\mathrm{tot}}\left(t\right)$ are the total strain, and $\varepsilon_{\mathrm{tot}}\left(t\right)$ is the light-driven contraction. For simplicity, the total strain is defined as $\varepsilon_{\mathrm{tot}}\left(t\right)=\lambda \left(X,t\right)-1$ [64], where λ(X,t) are written as
(3)$$\lambda_{1}(X_{1},t)=\frac{dx_{1}\left(X_{1},t\right)}{dX_{1}},\;\lambda_{2}(X_{2},t)=\frac{dx_{2}\left(X_{2},t\right)}{dX_{2}}.\;$$

Then, Equation (2) can be rewritten as
(4)$$\left\{\begin{array}{l} F_{f1}\left(t\right)=K_{f}L_{f}\left[\lambda_{1}\left(X_{1},t\right)-1-\varepsilon_{1}\left(X_{1},t\right)\right]\\ F_{f2}\left(t\right)=K_{f}L_{f}\left[\lambda_{2}\left(X_{2},t\right)-1-\varepsilon_{2}\left(X_{2},t\right)\right] \end{array}\right.,$$
where the light-driven contraction strain ε(X,t) is assumed to be proportional to the volume fraction of the isomers in the *cis* state φ(X,t) in the LCE fiber, which can be written as
(5)$$\left\{\begin{array}{l} \varepsilon_{1}\left(X_{1},t\right)=-C_{0}\varphi_{1}\left(X_{1},t\right)\\ \varepsilon_{2}\left(X_{2},t\right)=-C_{0}\varphi_{2}\left(X_{2},t\right) \end{array}\right.,$$
where $C_{0}$ is the contraction coefficient.

In order to find the instantaneous position $x_{1}$ ($x_{2}$) of the LCE fiber 1 (LCE fiber 2) at any time, we first rewrite $F_{f1}(t)$ ($F_{f2}(t)$) as $u_{1}\left(t\right)$ ($u_{2}\left(t\right)$) and $\varepsilon_{1}(X_{1},t)$ ($\varepsilon_{2}(X_{1},t)$). Considering that the LCE fiber is in uniaxial tensile state, the axial force is homogeneous although the contraction is inhomogeneous. Noting that $F_{f1}(t)$ and $F_{f2}(t)$ are homogeneous and constant in the LCE fibers, by integrating Equation (4) from 0 to $L_{f}$ on both sides, we can obtain
(6)$$\left\{\begin{array}{l} F_{f1}\left(t\right)=K_{f}\left[L_{f}\left(\lambda_{p}-1\right)+u_{1}(t)-\int_{0}^{L_{f}}\varepsilon_{1}\left(X_{1},t\right)dX_{1}\right]\\ F_{f2}\left(t\right)=K_{f}\left[L_{f}\left(\lambda_{p}-1\right)-u_{2}(t)-\int_{0}^{L_{f}}\varepsilon_{2}\left(X_{2},t\right)dX_{2}\right] \end{array}\right.,$$
where $\lambda_{p}$ is the pre-stretch of the LCE fiber.

Then, from Equation (2), $\lambda_{1}\left(X_{1},t\right)$, $\lambda_{2}\left(X_{2},t\right)$ can be expressed by $F_{f1}(t)$, $F_{f2}(t)$ as
(7)$$\left\{\begin{array}{l} \lambda_{1}\left(X_{1},t\right)=\frac{F_{f1}\left(t\right)}{K_{f}L_{f}}+1+\varepsilon_{1}\left(X_{1},t\right)\\ \lambda_{2}\left(X_{2},t\right)=\frac{F_{f2}\left(t\right)}{K_{f}L_{f}}+1+\varepsilon_{2}\left(X_{2},t\right) \end{array}\right..$$

By combining Equations (3) and (7), we can obtain
(8)$$\left\{\begin{array}{l} dx_{1}\left(X_{1},t\right)=\left[\frac{F_{f1}\left(t\right)}{K_{f}L_{f}}+1+\varepsilon_{1}\left(X_{1},t\right)\right]dX_{1}\\ dx_{2}\left(X_{2},t\right)=\left[\frac{F_{f1}\left(t\right)}{K_{f}L_{f}}+1+\varepsilon_{1}\left(X_{1},t\right)\right]dX_{2} \end{array}\right..\;$$

Combining Equations (6) and (8), we achieve
(9)$$\left\{\begin{array}{l} dx_{1}\left(X_{1},t\right)=\left[\varepsilon_{1}\left(X_{1},t\right)+\lambda_{p}+\frac{u_{1}(t)-\int_{0}^{L_{f}}\varepsilon_{1}\left(X_{1},t\right)dX_{1}}{L_{f}}\right]dX_{1}\\ dx_{2}\left(X_{2},t\right)=\left[\varepsilon_{1}\left(X_{1},t\right){+\lambda }_{p}-\frac{u_{2}(t)+\int_{0}^{L_{f}}\varepsilon_{2}\left(X_{2},t\right)dX_{2}}{L_{f}}\right]dX_{2} \end{array}\right..\;$$

By integrating both sides of the above formula from 0 to X, we achieve
(10)$$\left\{\begin{array}{l} x_{1}\left(X_{1},t\right)=\lambda_{p}X_{1}+\int_{0}^{X_{1}}\left[\varepsilon_{1}\left(X_{1},t\right)\right]dX_{1}+\frac{X_{1}}{L_{f}}\left[u_{1}\left(t\right)-\int_{0}^{L_{f}}\varepsilon_{1}\left(X_{1},t\right)dX_{1}\right]\\ x_{2}\left(X_{2},t\right)=\lambda_{p}X_{2}+\int_{0}^{X_{2}}\left[\varepsilon_{2}\left(X_{2},t\right)\right]dX_{2}-\frac{X_{2}}{L_{f}}\left[u_{2}\left(t\right)+\int_{0}^{L_{f}}\varepsilon_{2}\left(X_{2},t\right)dX_{2}\right] \end{array}\right..$$

The calculated $x_{1}$ and $x_{2}$ from Equation (10) can be compared to $\lambda_{p}L_{s}$ to determine whether the LCE fiber matter point is in the illuminated or non-illuminated area.

The two LCE fibers are coupled by the spring in the middle. Therefore, it is necessary for us to express the spring force $F_{s}\left(t\right)$. During the oscillation process, the elongation $\Delta L_{s}(t)$ of the spring at any moment can be expressed by the displacement of the two mass blocks $u_{1}(t)$ and $u_{2}(t)$ as
(11)$$\Delta L_{s}(t)=u_{2}(t)-u_{1}(t)+\frac{K_{f}L_{f}\left(\lambda_{p}-1\right)}{K_{s}}.\;$$

According to the deformation $\Delta L_{s}(t)$ of the spring and spring constant (coupling stiffness) $K_{s}$ of the spring, the spring force can be expressed as
(12)$$F_{s}(t)=K_{s}\left[u_{2}(t)-u_{1}(t)\right]+K_{f}L_{f}\left(\lambda_{p}-1\right).\;$$

### 2.2. Evolution Law of Number Fraction in the Two LCE Fibers

In order to calculate the light-driven contraction strain, we need to obtain the number fractions in the LCE fibers. According to the research of Yu et al., the *trans*-to-*cis* isomerization of LCE can be induced by UV or laser with a wavelength of less than 400 nm [62]. Generally, under UV light excitation, light-driven *cis*-to-*trans* isomerization can be neglected [65], and the number fraction of *cis*-isomers ϕ(t) depends on thermal excitation from *trans* to *cis*, thermally driven relaxation from *cis* to *trans*, and light-driven *trans*-to-*cis* isomerization. Considering that the thermal excitation from *trans* to *cis* is often negligible compared to the light-driven excitation, we define the number fraction of *cis* isomers in LCE fibers and use the following governing equations to describe the evolution of the number fraction of the *cis* isomers [66].
(13)$$\left\{\begin{array}{l} \frac{\partial \phi_{1}\left(X_{1},t\right)}{\partial t}=\eta_{0}I_{0}\left[1-\phi_{1}\left(X_{1},t\right)\right]-T_{0}^{-1}\phi_{1}\left(X_{1},t\right)\\ \frac{\partial \phi_{2}\left(X_{2},t\right)}{\partial t}=\eta_{0}I_{0}\left[1-\phi_{2}\left(X_{2},t\right)\right]-T_{0}^{-1}\phi_{2}\left(X_{2},t\right) \end{array}\right.,$$
where $T_{0}$ is the thermal relaxation time responding to the *cis* state to *trans* state, $I_{0}$ is the light intensity, and $\eta_{0}$ is a light-absorption constant.

### 2.3. Nondimensionalization

By defining the following dimensionless parameters: ${\tilde{F}}_{f1}\left(t\right)={F_{f1}\left(t\right)T_{0}^{2}/mL}_{f}$, ${\tilde{F}}_{f2}\left(t\right)={F_{f2}\left(t\right)T_{0}^{2}/mL}_{f}$, $\tilde{c}=cT_{0}^{}/m$, ${\tilde{F}}_{s}(t)={F_{s}\left(t\right)T_{0}^{2}/mL}_{f}$, ${\tilde{u}}_{1}(t)=u_{1}(t)/L_{{}_{f}}$, ${\tilde{u}}_{2}(t)=u_{2}(t)/L_{{}_{f}}$, $\tilde{t}=t/T_{0}$, ${\tilde{K}}_{f}=K_{f}T_{0}^{2}/m$, and ${\tilde{K}}_{s}=K_{s}T_{0}^{2}/m$, Equation (10) can be rewritten as
(14)$$\left\{\begin{array}{l} {\tilde{x}}_{1}\left({\tilde{X}}_{1},\tilde{t}\right)=\left[\lambda_{p}+{\tilde{u}}_{1}\left(\tilde{t}\right)\right]{\tilde{X}}_{1}-{\tilde{X}}_{1}\int_{0}^{1}\varepsilon_{1}\left({\tilde{X}}_{1},\tilde{t}\right)d{\tilde{X}}_{1}+\int_{0}^{{\tilde{X}}_{1}}\varepsilon_{1}\left({\tilde{X}}_{1},\tilde{t}\right)d{\tilde{X}}_{1}\\ {\tilde{x}}_{2}\left({\tilde{X}}_{2},\tilde{t}\right)=\left[\lambda_{p}-{\tilde{u}}_{2}\left(\tilde{t}\right)\right]{\tilde{X}}_{2}-{\tilde{X}}_{2}\int_{0}^{1}\varepsilon_{2}\left({\tilde{X}}_{2},\tilde{t}\right)d{\tilde{X}}_{2}+\int_{0}^{{\tilde{X}}_{2}}\varepsilon_{2}\left({\tilde{X}}_{2},\tilde{t}\right)d{\tilde{X}}_{2} \end{array}\right..\;$$

Combining Equations (1), (6), and (12), we achieve
(15)$$\left\{\begin{array}{l} {\tilde{\ddot{u}}}_{1}={\tilde{K}}_{s}\left[{\tilde{u}}_{2}(t)-{\tilde{u}}_{1}(t)\right]-{\tilde{K}}_{f}\left[{\tilde{u}}_{1}(\tilde{t})-\int_{0}^{1}\varepsilon_{1}\left({\tilde{X}}_{1},\tilde{t}\right)d{\tilde{X}}_{1}\right]-c{\tilde{\dot{u}}}_{1}\\ {\tilde{\ddot{u}}}_{2}=-{\tilde{K}}_{f}\left[{\tilde{u}}_{2}(\tilde{t})+\int_{0}^{1}\varepsilon_{1}\left({\tilde{X}}_{1},\tilde{t}\right)d{\tilde{X}}_{1}\right]-{\tilde{K}}_{s}\left[{\tilde{u}}_{2}(\tilde{t})-{\tilde{u}}_{1}(t)\right]-c{\tilde{\dot{u}}}_{2} \end{array}\right.,$$
where $\tilde{\dot{u}}$ and $\tilde{\ddot{u}}$ indicate the velocity $\frac{d\tilde{u}\left(\tilde{t}\right)}{d\tilde{t}}$ and acceleration $\frac{d^{2}\tilde{u}\left(\tilde{t}\right)}{dt^{2}}$ of the mass, respectively.

By defining the parameter $\tilde{I}=T_{0}\eta_{0}I_{0}$, Equation (13) can be rewritten as
(16)$$\left\{\begin{array}{l} \frac{\partial \phi_{1}\left({\tilde{X}}_{1},\tilde{t}\right)}{\partial \tilde{t}}=\tilde{I}-\left(1+\tilde{I}\right)\phi_{1}\left({\tilde{X}}_{1},\tilde{t}\right)\\ \frac{\partial \phi_{2}\left({\tilde{X}}_{2},\tilde{t}\right)}{\partial \tilde{t}}=\tilde{I}-\left(1+\tilde{I}\right)\phi_{2}\left({\tilde{X}}_{2},\tilde{t}\right) \end{array}\right..\;$$

### 2.4. Solution Method

In this paper, the LCE fiber is discretized into J material points (let J=500 for both LCE fibers in this paper) in the numerical calculation process. The position vector of each material point of LCE fiber 1 and LCE fiber 2 in the Lagrangian coordinate system can be expressed as $X_{1}=\left(X_{1}^{1},X_{1}^{2}\cdot \cdot \cdot X_{1}^{J}\right)$, $X_{2}=\left(X_{2}^{1},X_{2}^{2}\cdot \cdot \cdot X_{2}^{J}\right)$, and is called the material coordinate. The position vector of each material point of LCE fiber 1 and LCE fiber 2 in the Eulerian coordinate system can be expressed as $x_{1}=\left(x_{1}^{1},x_{1}^{2}\cdot \cdot \cdot x_{1}^{J}\right)$, $x_{2}=\left(x_{2}^{1},x_{2}^{2}\cdot \cdot \cdot x_{2}^{J}\right)$, which is called the spatial coordinate. By using the different methods to solve the dimensionless equation in Equation (16), the expression for the *cis* number fraction at any moment in the LCE fiber can be written as
(17)$$\left\{\begin{array}{l} \phi_{1}^{{}_{n+1}}=\phi_{1}^{{}_{n}}+\left[\tilde{I}-(1+\tilde{I})\phi_{1}^{{}_{n}}\right]\Delta \tilde{t}\\ \phi_{2}^{{}_{n+1}}=\phi_{2}^{{}_{n}}+\left[\tilde{I}-(1+\tilde{I})\phi_{2}^{{}_{n}}\right]\Delta \tilde{t} \end{array}\right..\;$$

Substituting Equation (17) into Equation (4), we are able to obtain the light-driven contraction strain $\varepsilon_{11}$, $\varepsilon_{21}$ for LCE fiber 1 and LCE fiber 2 located in the illuminated area
(18)$$\left\{\begin{array}{l} \varepsilon_{11}=-C_{0}\left\{\phi_{1}^{n}+\left[\tilde{I}-\left(1+\tilde{I}\right)\phi_{1}^{n}\right]\Delta \tilde{t}\right\}\\ \varepsilon_{21}=-C_{0}\left\{\phi_{2}^{n}+\left[\tilde{I}-\left(1+\tilde{I}\right)\phi_{2}^{n}\right]\Delta \tilde{t}\right\} \end{array}\right..\;$$

Similarly, when the LCE fibers are located in a non-illuminated area, the shrinkage strain $\varepsilon_{12}$, $\varepsilon_{22}$ can be written as
(19)$$\left\{\begin{array}{l} \varepsilon_{12}=-C_{0}\left(\phi_{1}^{n}-\phi_{1}^{n}\Delta \tilde{t}\right)\\ \varepsilon_{22}=-C_{0}\left(\phi_{2}^{n}-\phi_{2}^{n}\Delta \tilde{t}\right) \end{array}\right..\;$$

Equation (15) is an ordinary differential equation with variable coefficients of second order, so no analytical solution can be obtained. Herein, we used the classical fourth-order Runge–Kutta method to solve the differential equation using *Matlab* software. By iterating Equations (15) and (17)–(19) we are able to obtain the final steady-state response of the LCE spring oscillator, i.e., the relationship between displacement and velocity with time histories.

## 3. Three Synchronization Modes and Their Mechanisms

### 3.1. Three Synchronization Modes

To investigate the collective motion of the two spring oscillators, we first need to estimate the typical values of the dimensionless parameters in the model. From the accessible experiments [19,49,66,67], the typical values of the material properties and geometric parameters are listed in Table 1.

Figure 2 shows three steady synchronization modes of the self-oscillation coupling system. In the computation, we fix ${\tilde{K}}_{f}=5$, $\tilde{c}=0.1$, $\lambda_{p}=1.15$, $\tilde{I}=0.5$, $C_{0}=0.7$, ${\tilde{u}}_{1}^{0}=0.2$, ${\tilde{u}}_{2}^{0}=0$, ${\tilde{\dot{u}}}_{1}^{0}=0.3$, and ${\tilde{\dot{u}}}_{2}^{0}=-0.3$. For ${\tilde{K}}_{s}=0.05$; the result shows that the two LCE oscillators oscillate the anti-phase, and the time–history curve and domain of attraction of the anti-phase mode are given in Figure 2b,c, respectively. For ${\tilde{K}}_{s}=0.15$, the results show that the amplitude of the two LCE oscillators changes periodically, which means that the self-oscillation coupling system is in non-phase-locked mode, as shown in Figure 2d,e. Figure 2f is the domain of attraction in the non-phase-locked mode. For ${\tilde{K}}_{s}=0.5$, the calculation shows that the two curves of the two LCE oscillators coincide, which means that the two LCE oscillators are in in-phase mode, as show in Figure 2g,h. Figure 2i is the domain of attraction of ${\tilde{u}}_{1}$ and ${\tilde{u}}_{2}$ in in-phase mode. When other physical parameters remain unchanged, the conversion of different synchronization modes can be realized by changing one of the parameters, such as by changing the coupling stiffness ${\tilde{K}}_{s}$.

### 3.2. The Mechanism of Self-Excited Oscillation

To investigate the mechanism of the self-excited oscillation of the LCE oscillator under uniform and constant illumination, Figure 3 plots the mechanism of the self-excited oscillation in anti-phase mode for ${\tilde{K}}_{f}=5$,${\tilde{K}}_{s}=0.1$, $\tilde{c}=0.1$, $\lambda_{p}=1.15$, $\tilde{I}=0.5$, $C_{0}=0.7$, ${\tilde{u}}_{1}^{0}=0$, ${\tilde{u}}_{2}^{0}=0$, ${\tilde{\dot{u}}}_{1}^{0}=0.4$, and ${\tilde{\dot{u}}}_{2}^{0}=-0.4$. During self-excited oscillation, the number fraction $\phi_{1}\left(\phi_{2}\right)$ of *cis*-isomers, the shrinkage strain $\varepsilon_{1}\left(\varepsilon_{2}\right)$, the driving force ${\tilde{F}}_{f1}\left({\tilde{F}}_{f2}\right)$, the spring force ${\tilde{F}}_{s}$, and the displacement ${\tilde{u}}_{1}\left({\tilde{u}}_{2}\right)$ of mass block change periodically with the different time histories. In Figure 3d, the driving force and the displacement of the mass show a closed-loop relationship, and the area enclosed by the closed loop represents the network created by the driving force, which compensates the energy loss of the system and maintains the periodic self-excited oscillation of the system. Figure 3f represents the dependence of the displacement on time. It can be seen from this figure that the two mass blocks have a phase difference of half a cycle.

Numerical calculations show that the mechanisms of the self-excited oscillation of in-phase mode and non-phase-locked mode are similar to that of anti-phase mode. The number fraction of *cis*-isomers, the shrinkage strain, the driving force, the spring force, and the displacement of two mass blocks change periodically with time histories too. Driving force and mass displacement are also in a closed-loop relationship, and the area enclosed by the closed loop represents the network created by the driving force. This is because a part of the LCE fiber continuously enters and exits the illuminated area, causing the periodic contraction and relaxation of the LCE fiber, which causes the self-excited oscillation of the two mass blocks.

### 3.3. Triggering Conditions for Three Synchronization Modes

Numerical calculations show that the synchronization mode depends on the combination of various parameters. We took the combination of coupling stiffness and initial state as an example to study the effects of the system parameters on synchronous mode. In the computation, we fixed ${\tilde{K}}_{f}=5$, $\tilde{c}=0.1$, $\lambda_{p}=1.15$, $\tilde{I}=0.5$, $C_{0}=0.7$, ${\tilde{u}}_{1}^{0}=0$, ${\tilde{u}}_{2}^{0}=0$, and ${\tilde{\dot{u}}}_{1}^{0}=-0.3$ and studied the effects of the phase difference between the two mass blocks and the coupling stiffness on the three synchronization modes at the initial moment by changing ${\tilde{\dot{u}}}_{2}^{0}$ and ${\tilde{K}}_{s}$. By analyzing a large amount of calculation data, the attractive domains of the three steady synchronization modes can be drawn, as shown in Figure 4.

Figure 4 shows the phase diagram of three steady synchronization modes under different coupling stiffnesses and different initial states. It can be seen from the figure that the smaller the coupling stiffness and the greater the phase difference between the two mass blocks of the initial state are, the easier it is to obtain the anti-phase mode. Conversely, it is conducive to the realization of the in-phase mode. In general, the in-phase mode is more easily achieved than the anti-phase mode, indicating a difference in robustness and the corresponding basins of attraction between the three synchronization modes. When the phase difference between the two LCE spring oscillators corresponding to the initial condition is large enough and the coupling stiffness ${\tilde{K}}_{s}$ is of moderate size, then there is a stable non-phase-locked mode between the in-phase mode and anti-phase mode. It is worth mentioning that when the two mass blocks are in anti-phase at the initial state (i.e., ${\tilde{\dot{u}}}_{1}^{0}=0.3$), then the phase diagram of the anti-phase synchronization mode is the largest. The system is in anti-phase mode when ${\tilde{K}}_{s}<2.9$, and when ${\tilde{K}}_{s}\geq 2.9$, it is in in-phase mode.

After sorting out a large amount of calculation data, we briefly summarize the influence of other parameters on the three steady synchronization modes. When other parameters remain unchanged and the smaller the light intensity and contraction coefficient are, the easier it is to obtain the anti-phase mode, and the larger the damping coefficient and the spring coefficient of the LCE fiber are, the easier it is to obtain the anti-phase mode. Conversely, it is conducive to the realization of the in-phase mode. When the size of other parameters is moderate, there is a stable non-phase-locked mode between the in-phase mode and anti-phase mode. Studying the phase diagram of three steady synchronization modes under different parameters is extremely important in the design of automatic machines and soft robots.

## 4. Parametric Study

### 4.1. Effect of Coupling Stiffness

Figure 5a,b plot the effect of three different coupling stiffnesses on the self-oscillation model in the in-phase mode. The parameters are ${\tilde{K}}_{f}=5$, $\tilde{c}=0.1$, $\lambda_{p}=1.15$, $\tilde{I}=0.5$, $C_{0}=0.7$, ${\tilde{u}}_{1}^{0}=0$, ${\tilde{u}}_{2}^{0}=0$, ${\tilde{\dot{u}}}_{1}^{0}=-0.3$, and ${\tilde{\dot{u}}}_{2}^{0}=-0.2$. Under the action of different coupling stiffness, the domain of attraction ${\tilde{u}}_{1}\left(t\right)$ and ${\tilde{u}}_{2}\left(t\right)$ overlap, and the corresponding limit cycles also overlap, indicating that the coupling stiffness ${\tilde{K}}_{s}$ has no effect on self-oscillation in in-phase mode. This is because when the LCE oscillators are in in-phase mode, the distance between the two mass blocks remains unchanged, and the spring force is a constant force. Figure 5c,d plot the effect of three different coupling stiffnesses on the self-oscillation in the anti-phase mode. The parameters are ${\tilde{K}}_{f}=5$, $\tilde{c}=0.1$, $\lambda_{p}=1.15$, $\tilde{I}=0.65$, $C_{0}=0.7$, ${\tilde{u}}_{1}^{0}=0$, ${\tilde{u}}_{2}^{0}=0$, ${\tilde{\dot{u}}}_{1}^{0}=-0.3$, and ${\tilde{\dot{u}}}_{2}^{0}=0.3$. The results show that the larger the coupling stiffness, the smaller the limit cycle and the stronger the suppression of self-excited oscillation. The reason for this phenomenon is because the light energy absorbed from the environment decreases as the coupling stiffness ${\tilde{K}}_{s}$ increases in anti-phase mode. Figure 5e–h plot the effect of coupling stiffness on the non-phase-locked mode for ${\tilde{K}}_{f}=5$, $\tilde{c}=0.1$, $\lambda_{p}=1.15$, $\tilde{I}=0.5$, $C_{0}=0.7$, ${\tilde{u}}_{1}^{0}=0$, ${\tilde{u}}_{2}^{0}=0$, ${\tilde{\dot{u}}}_{1}^{0}=-0.3$, and ${\tilde{\dot{u}}}_{2}^{0}=0.25$. The results are similar to the anti-phase mode; that is, the larger the coupling stiffness, the smaller the limit cycle and the stronger the suppression of self-excited oscillation. The change of the amplitude within a doubling period increases as the coupling stiffness ${\tilde{K}}_{s}$ increases within a doubling period. More calculations show that the effects of the coupling stiffness are the same in non-phase-locked mode. By calculating the data, it was determined that the self-oscillation frequencies in the anti-phase mode and non-phase-locked mode increase as the coupling stiffness ${\tilde{K}}_{s}$ increases.

### 4.2. Effect of Light Intensity

Figure 6a,b plot the effect of three different light intensities on self-oscillation in in-phase mode. The parameters are ${\tilde{K}}_{f}=5$, $\tilde{c}=0.1$, $\lambda_{p}=1.15$, ${\tilde{K}}_{s}=0.3$, $C_{0}=0.7$, ${\tilde{u}}_{1}^{0}=0$, ${\tilde{u}}_{2}^{0}=0$, ${\tilde{\dot{u}}}_{1}^{0}=-0.3$, and ${\tilde{\dot{u}}}_{1}^{0}=-0.2$. Figure 6c,d plot the effect of three different light intensities on self-oscillation in anti-phase mode. The parameters are ${\tilde{K}}_{f}=5$, $\tilde{c}=0.1$, $\lambda_{p}=1.15$, ${\tilde{K}}_{s}=0.1$, $C_{0}=0.7$, ${\tilde{u}}_{1}^{0}=0$, ${\tilde{u}}_{2}^{0}=0$, ${\tilde{\dot{u}}}_{1}^{0}=-0.3$, and ${\tilde{\dot{u}}}_{1}^{0}=0.3$. The light intensities have the same effect on the in-phase mode and the anti-phase mode. The results show that the smaller the light intensity, the smaller the limit cycle, and the stronger the suppression of self-excited oscillation, as shown in Figure 6b,d. This is because the light-driven contraction of the LCE fiber increases as the light intensity increases, and the light energy absorbed by the LCE fiber from the environment increases. This is consistent with the self-excited oscillation of a single oscillator [55]. Figure 6e–h plot the effect of light intensity on the non-phase-locked mode for ${\tilde{K}}_{f}=5$, $\tilde{c}=0.1$, $\lambda_{p}=1.15$, ${\tilde{K}}_{s}=0.2$, $C_{0}=0.6$, ${\tilde{u}}_{1}^{0}=0$, ${\tilde{u}}_{2}^{0}=0$, ${\tilde{\dot{u}}}_{1}^{0}=-0.3$, and ${\tilde{\dot{u}}}_{1}^{0}=0.1$. It can be seen that the greater the light intensity, the greater the amplitude and the corresponding limit cycle and the greater the change of the amplitude within a doubling period, until it evolves into in-phase mode. The results show that the light intensity has no effect on the frequency of the three synchronization modes.

### 4.3. Effect of Contraction Coefficient

Figure 7a,b plot the effect of three different contraction coefficients on the self-oscillation in the in-phase mode. The parameters are ${\tilde{K}}_{f}=5$, $\tilde{c}=0.1$, $\lambda_{p}=1.15$, ${\tilde{K}}_{s}=0.3$, ${\tilde{u}}_{1}^{0}=0$, ${\tilde{u}}_{2}^{0}=0$, ${\tilde{\dot{u}}}_{1}^{0}=-0.3$, and ${\tilde{\dot{u}}}_{1}^{0}=-0.2$. Figure 7c,d plot the effect of three different contraction coefficients on the self-oscillation in the anti-phase mode. The parameters are ${\tilde{K}}_{f}=5$, $\tilde{c}=0.1$, $\lambda_{p}=1.15$, ${\tilde{K}}_{s}=0.1$, ${\tilde{u}}_{1}^{0}=0$, ${\tilde{u}}_{2}^{0}=0$, ${\tilde{\dot{u}}}_{1}^{0}=-0.3$, and ${\tilde{\dot{u}}}_{1}^{0}=0.3$. The contraction coefficients have the same effect on the in-phase mode and the anti-phase mode. It can be seen form Figure 7a,c that the amplitudes increase as the contraction coefficient increases. The smaller the contraction coefficient and the smaller the corresponding limit cycle are shown in Figure 7b,d. The reason for this phenomenon is that the light-driven contraction of the LCE fiber increases as the contraction coefficient $C_{0}$ increases and as the light energy that is absorbed from the environment increases. This is consistent with the self-excited oscillation of a single oscillator [55]. Figure 7e–h plot the effect of the contraction coefficient on the non-phase-locked mode for ${\tilde{K}}_{f}=5$, $\tilde{c}=0.1$, $\lambda_{p}=1.15$, ${\tilde{K}}_{s}=0.2$, $\tilde{I}=0.6$, ${\tilde{u}}_{1}^{0}=0$, ${\tilde{u}}_{2}^{0}=0$, ${\tilde{\dot{u}}}_{1}^{0}=-0.3$, and ${\tilde{\dot{u}}}_{1}^{0}=0.1$. It can be seen from the figures that the greater the contraction coefficient, the greater the amplitude and the corresponding limit cycle, and the greater the change of the amplitude within a doubling period, until it evolves into in-phase mode. More calculations show that the effect of the contraction coefficient is the same during non-phase-locked mode and that it has no effect on the frequency of the three synchronization modes.

### 4.4. Effect of Damping Coefficient

Figure 8a,b plot the effect of three different damping coefficients on self-oscillation in the in-phase mode. The parameters are ${\tilde{K}}_{f}=5$, $\lambda_{p}=1.15$, ${\tilde{K}}_{s}=0.2$, $\tilde{I}=0.6$, $C_{0}=0.7$, ${\tilde{u}}_{1}^{0}=0$, ${\tilde{u}}_{2}^{0}=0$, ${\tilde{\dot{u}}}_{1}^{0}=-0.3$, and ${\tilde{\dot{u}}}_{1}^{0}=-0.2$. Figure 8c,d plot the effect of three different damping coefficients on self-oscillation in the anti-phase mode. The parameters are ${\tilde{K}}_{f}=5$, $\lambda_{p}=1.15$, ${\tilde{K}}_{s}=0.2$, $\tilde{I}=0.8$, $C_{0}=0.7$, ${\tilde{u}}_{1}^{0}=0$, ${\tilde{u}}_{2}^{0}=0$, ${\tilde{\dot{u}}}_{1}^{0}=-0.3$, and ${\tilde{\dot{u}}}_{1}^{0}=0.3$. The damping coefficients have the same effect on the in-phase mode and the anti-phase mode. The results show that the greater the damping coefficient, the smaller the corresponding limit cycle, as shown in Figure 8b,d. This is because that as the damping coefficient increases, the energy consumed by the system increases, and the net energy input during the oscillating decreases. This is consistent with physical tuition and other self-oscillating systems [57]. Figure 8e–h plot the effect of the damping coefficient on self-oscillation in non-phase-locked mode. The parameters are ${\tilde{K}}_{f}=5$, $\lambda_{p}=1.15$, ${\tilde{K}}_{s}=0.2$, $\tilde{I}=0.8$, $C_{0}=0.7$, ${\tilde{u}}_{1}^{0}=0$, ${\tilde{u}}_{2}^{0}=0$, ${\tilde{\dot{u}}}_{1}^{0}=-0.3$, and ${\tilde{\dot{u}}}_{1}^{0}=0.1$. It can be seen from the figure that the greater the damping coefficient, the smaller the amplitude, limit cycle, and the change of the amplitude, and it then evolves into anti-phase mode until it is in static mode. More calculations show that the frequency of the three synchronization modes does not change with the changes in the damping coefficient.

### 4.5. Effect of Spring Constant of LCE Fiber

Figure 9a,b plot the effect of three different spring constants of the LCE fiber on the self-oscillation in in-phase mode. The parameters are $\tilde{c}=0.1$, $\lambda_{p}=1.15$, ${\tilde{K}}_{s}=0.3$, $\tilde{I}=0.5$, $C_{0}=0.7$, ${\tilde{u}}_{1}^{0}=0$, ${\tilde{u}}_{2}^{0}=0$, ${\tilde{\dot{u}}}_{1}^{0}=-0.3$, and ${\tilde{\dot{u}}}_{1}^{0}=-0.1$. Figure 9c,d plot the effect of three different spring constants on self-oscillation in anti-phase mode. The parameters are $\tilde{c}=0.1$, $\lambda_{p}=1.15$, ${\tilde{K}}_{s}=0.3$, $\tilde{I}=0.6$, $C_{0}=0.7$, ${\tilde{u}}_{1}^{0}=0$, ${\tilde{u}}_{2}^{0}=0$, ${\tilde{\dot{u}}}_{1}^{0}=-0.3$, and ${\tilde{\dot{u}}}_{1}^{0}=0.3$. The spring constants of LCE fiber ${\tilde{K}}_{f}$ have the same effect on in-phase mode and anti-phase mode. The results show that the greater the spring constant of the LCE fiber, the greater the corresponding limit cycle is, as shown in Figure 9b,d. The reason for this phenomenon is that the light energy that is absorbed from the environment increases as the spring constant of the LCE fiber ${\tilde{K}}_{f}$ increases. This is consistent with the self-excited oscillation of a single oscillator [56]. Figure 9e–h plot the effect of the spring constant of the LCE fiber on self-oscillation in non-phase-locked mode. The parameters are $\tilde{c}=0.1$, $\lambda_{p}=1.15$, ${\tilde{K}}_{s}=0.3$, $\tilde{I}=0.5$, $C_{0}=0.7$, ${\tilde{u}}_{1}^{0}=0$, ${\tilde{u}}_{2}^{0}=0$, ${\tilde{\dot{u}}}_{1}^{0}=-0.3$, and ${\tilde{\dot{u}}}_{1}^{0}=0.2$. It can be seen from the figure that the greater the spring constant of the LCE fiber, the greater the amplitude, the limit cycle, and the change in the amplitude within a doubling period, and it then evolves into anti-phase mode. This is because as the spring constant of the LCE fiber ${\tilde{K}}_{f}$ increases, the ratio of the spring force ${\tilde{F}}_{s}$ to the driving force ${\tilde{F}}_{f1}\left({\tilde{F}}_{f2}\right)$ decreases, which is equivalent to reducing the coupling stiffness ${\tilde{K}}_{s}$ and is more conducive to the realization of anti-phase mode, and this result corresponds to Figure 4. More calculations show that the self-oscillation frequencies of the three synchronization modes increase as the spring constant of the LCE fiber ${\tilde{K}}_{f}$ increases.

### 4.6. Effect of Initial Condition

Figure 10a,b plot the effect of the initial condition on self-oscillation in in-phase mode for ${\tilde{K}}_{f}=5$, $\tilde{c}=0.1$, $\lambda_{p}=1.15$, ${\tilde{K}}_{s}=0.2$, $\tilde{I}=0.5$, $C_{0}=0.7$, ${\tilde{u}}_{1}^{0}=0$, ${\tilde{u}}_{2}^{0}=0$, and ${\tilde{\dot{u}}}_{1}^{0}=-0.3$. Figure 10c,d plot the effect of the initial condition on self-oscillation in anti-phase mode for ${\tilde{K}}_{f}=5$, $\tilde{c}=0.1$, $\lambda_{p}=1.15$, ${\tilde{K}}_{s}=0.05$, $\tilde{I}=0.65$, $C_{0}=0.7$, ${\tilde{u}}_{1}^{0}=0$, ${\tilde{u}}_{2}^{0}=0$, ${\tilde{\dot{u}}}_{1}^{0}=-0.3$, and ${\tilde{\dot{u}}}_{2}^{0}=0.3$. Under the action of the different initial conditions, the domains of attraction ${\tilde{u}}_{1}\left(t\right)$ and ${\tilde{u}}_{2}\left(t\right)$ overlap, and the corresponding limit cycles also overlap, indicating that the initial conditions have no effect on the self-excited oscillation in in-phase mode and anti-phase mode. This is because the initial condition does not affect the elastic strain energy input to the system. More calculations also show that the influence of initial condition is the same, and this is consistent with the self-excited oscillation of a single oscillator [55,60]. Figure 10e–h plot the effect of initial condition on the self-oscillation in non-phase-locked mode for ${\tilde{K}}_{f}=5$, $\tilde{c}=0.1$, $\lambda_{p}=1.15$, ${\tilde{K}}_{s}=0.2$, $\tilde{I}=0.5$, $C_{0}=0.7$, ${\tilde{u}}_{1}^{0}=0$, ${\tilde{u}}_{2}^{0}=0$, and ${\tilde{\dot{u}}}_{1}^{0}=-0.3$. It can be seen from the figures that the smaller the phase difference between the two mass blocks of the initial state, the greater the amplitude of the amplitude change until it evolves into in-phase mode. More calculations show that the initial conditions have no effect on the frequency of the three synchronization modes.

## 5. Equivalent Systems

Figure 11a,b plot the equivalent systems of the in-phase mode and anti-phase mode, respectively. In in-phase mode, the length of the spring does not change with time, and thus, the system in in-phase mode is equivalent to a single self-excited oscillator with a constant spring force (Figure 11a). Furthermore, it can be easily predicted that the amplitude of the equivalent system increases as the light intensity, contraction coefficient, and spring coefficient of the LCE fiber, and decreases as the damping coefficient increases. The frequency of the self-excited oscillation increases as the spring coefficient of the LCE fiber increases.

In the anti-phase mode, the midpoint of the spring does not move during the oscillation, and thus, the system is equivalent to single self-excited oscillator constrained by a fixed spring with half of its original length, as shown in Figure 11b. Therefore, it can also be predicted that the amplitude of the equivalent system increases as the light intensity, contraction coefficient, and spring coefficient of the LCE fiber increases and decreases as the damping coefficient and spring coefficient of the spring increase. The frequency of self-excited oscillation increases as the spring coefficient of the LCE fiber and spring coefficient of the spring increase.

## 6. Conclusions

The study of the self-excited oscillation coupling of two or more systems and their collective motion is beneficial to the construction of richer and more complex motions, and the more abundant motion modes are constructed, the richer the functions of micro-robots will be. In this article, we constructed two joint LCE spring oscillators that are connected by a spring and theoretically investigated their synchronization phenomenon based on the well-established dynamic LCE model. The numerical calculations show that the self-oscillation coupling system has three steady synchronization modes: in-phase mode, anti-phase mode, and non-phase-locked mode. The synchronization mode depends on the combination of various parameters, and we provided phase diagrams for three steady synchronization modes under different coupling stiffnesses and different initial states. In addition, the results show that the amplitude of three synchronization modes and the changes in the amplitude within a doubling period in the non-phase-locked mode increased as the light intensity, contraction coefficient, and spring coefficient of the LCE fiber increased and decreased as the damping coefficient decreased. The frequencies of the anti-phase mode and non-phase-locked mode increased as the coupling stiffness and spring coefficients of the LCE fiber increased, and the frequency of the in-phase mode increased as the spring coefficients of the LCE fiber increased. The coupling stiffness and initial conditions had no effect on the in-phase mode. These results are consistent with the self-excited oscillation of a single oscillator, for which we also proposed equivalent systems in the in-phase mode and the anti-phase mode. The coupled self-oscillation system proposed in this paper may be used to design complex systems and have broad application prospects in the fields of signal monitoring, medical equipment, micro-active machines, and micro-robots.

## Figures and Tables

**Figure 1 micromachines-13-00271-f001:**
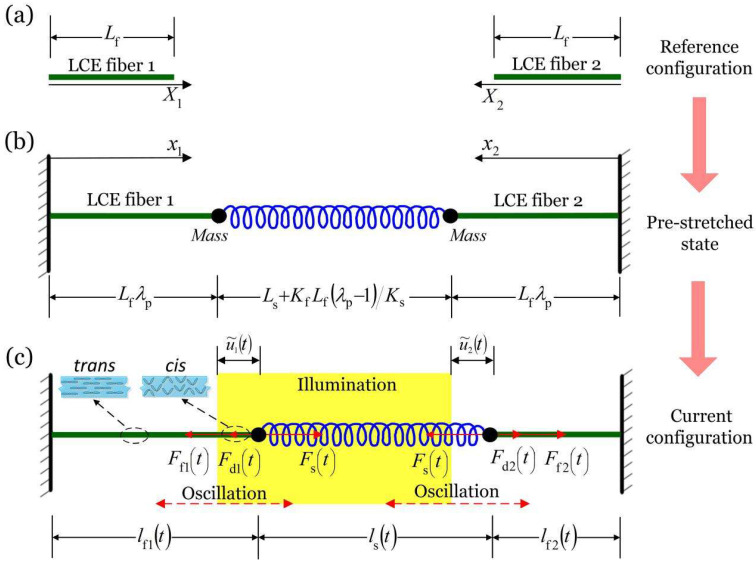
The dynamic model of the self-oscillation coupling system is composed of two identical LCE fibers and a spring. (**a**) Reference configuration. (**b**) Pre-stretched state. (**c**) Current configuration. Under uniform and constant illumination, under uniform and constant illumination, the self-oscillation of the spring oscillators can be triggered by the coupling between the light-driven contraction of the fibers and their movement.

**Figure 2 micromachines-13-00271-f002:**
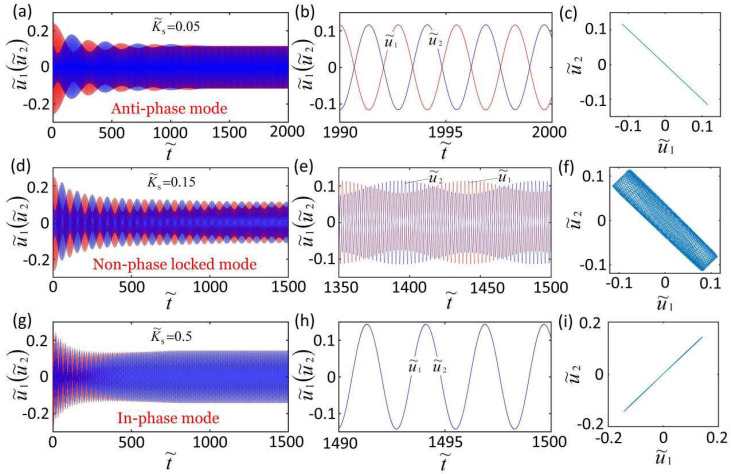
Three steady synchronization modes of the self-oscillation coupling system. The other parameters are ${\tilde{K}}_{f}=5$, $\tilde{c}=0.1$, $\lambda_{p}=1.15$, $\tilde{I}=0.5$, $C_{0}=0.7$, ${\tilde{u}}_{1}^{0}=0.2$, ${\tilde{u}}_{2}^{0}=0$, ${\tilde{\dot{u}}}_{1}^{0}=0.3$, and ${\tilde{\dot{u}}}_{2}^{0}=-0.3$. (**a**–**c**) Anti-phase synchronization mode (${\tilde{K}}_{s}=0.05$ ). (**d**–**f**) Non-phase-locked mode (${\tilde{K}}_{s}=0.15$ ). (**g**–**i**) In-phase synchronization mode (${\tilde{K}}_{s}=0.5$ ).

**Figure 3 micromachines-13-00271-f003:**
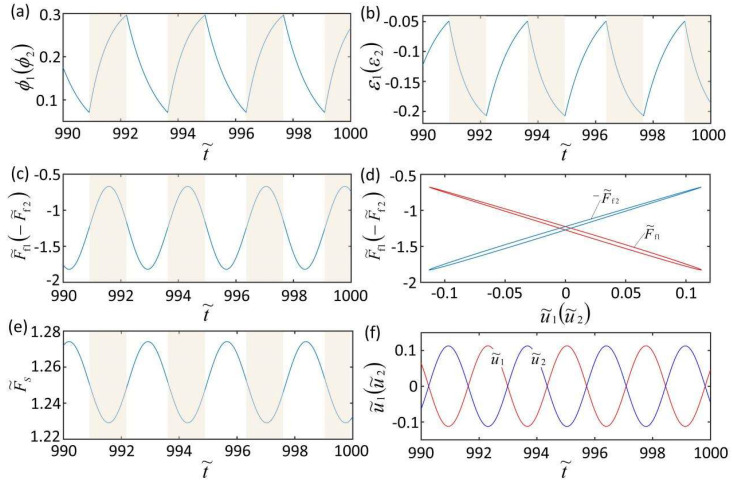
Mechanism of the self-excited oscillation in anti-phase mode. The parameters are ${\tilde{K}}_{f}=5$, ${\tilde{K}}_{s}=0.1$, $\tilde{c}=0.1$, $\lambda_{p}=1.15$, $\tilde{I}=0.5$, $C_{0}=0.7$, ${\tilde{u}}_{1}^{0}=0$, ${\tilde{u}}_{2}^{0}=0$, ${\tilde{\dot{u}}}_{1}^{0}=0.3$, and ${\tilde{\dot{u}}}_{2}^{0}=-0.3$. The illuminated zone is represented by the shaded area. (**a**) Time histories of the number fractions of cis-isomers in the two LCE fibers. (**b**) Time histories of contraction strains. (**c**) The spring forces of the two LCE fibers (driving forces). (**d**) The dependence of the spring force of LCE fiber on the mass displacement. (**e**) The spring force of the spring (spring force); (**f**) The displacements of the two mass blocks.

**Figure 4 micromachines-13-00271-f004:**
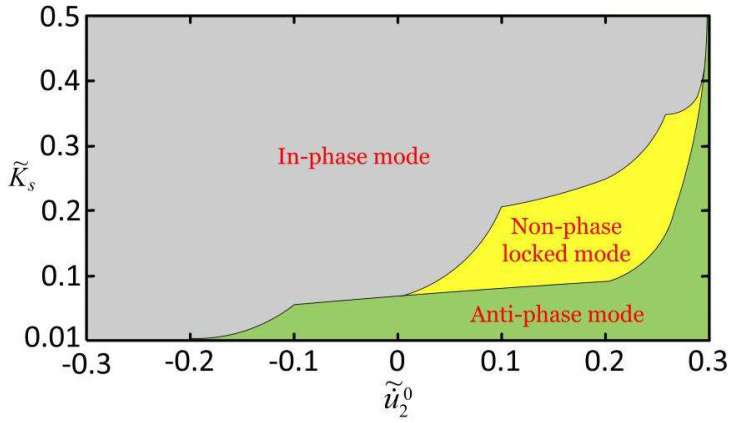
Dependence of the synchronization mode on the coupling stiffness and initial state. The parameters are ${\tilde{K}}_{f}=5$, $\tilde{c}=0.1$, $\lambda_{p}=1.15$, $\tilde{I}=0.5$, $C_{0}=0.7$, ${\tilde{u}}_{1}^{0}=0$, ${\tilde{u}}_{2}^{0}=0$, and ${\tilde{\dot{u}}}_{1}^{0}=-0.3$. Where the purple area is the phase diagram of the in-phase mode, the yellow area is the phase diagram of the anti-phase mode, and the green area is the phase diagram of the anti-phase mode.

**Figure 5 micromachines-13-00271-f005:**
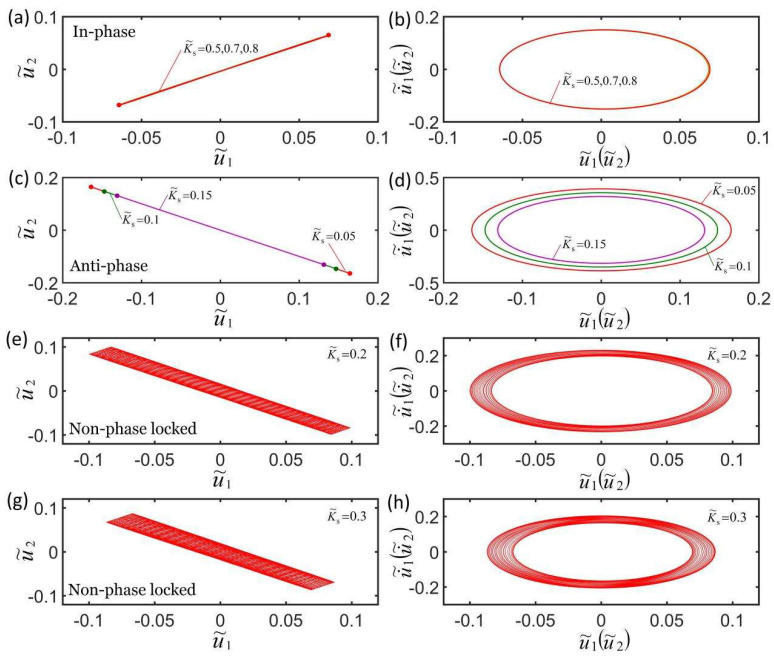
Effect of coupling stiffness on the self-oscillation for a constant synchronous mode. (**a**,**b**) Effect of coupling stiffness on the in-phase mode for ${\tilde{K}}_{f}=5$, $\tilde{c}=0.1$, $\lambda_{p}=1.15$, $\tilde{I}=0.5$, $C_{0}=0.7$, ${\tilde{u}}_{1}^{0}=0$, ${\tilde{u}}_{2}^{0}=0$, ${\tilde{\dot{u}}}_{1}^{0}=-0.3$, and ${\tilde{\dot{u}}}_{2}^{0}=-0.2$. (**c**,**d**) Effect of coupling stiffness on the anti-phase mode for ${\tilde{K}}_{f}=5$, $\tilde{c}=0.1$, $\lambda_{p}=1.15$, $\tilde{I}=0.65$, $C_{0}=0.7$, ${\tilde{u}}_{1}^{0}=0$, ${\tilde{u}}_{2}^{0}=0$, ${\tilde{\dot{u}}}_{1}^{0}=-0.3$, and ${\tilde{\dot{u}}}_{2}^{0}=0.3$. (**e**–**h**) Effect of coupling stiffness on the non-phase-locked mode for ${\tilde{K}}_{f}=5$, $\tilde{c}=0.1$, $\lambda_{p}=1.15$, $\tilde{I}=0.5$, $C_{0}=0.7$, ${\tilde{u}}_{1}^{0}=0$, ${\tilde{u}}_{2}^{0}=0$, ${\tilde{\dot{u}}}_{1}^{0}=-0.3$, and ${\tilde{\dot{u}}}_{2}^{0}=0.25$. The coupling stiffness has no effect on the self-oscillation in in-phase mode and has an effect on the period and amplitude of the self-oscillation in anti-phase mode and non-phase-locked mode.

**Figure 6 micromachines-13-00271-f006:**
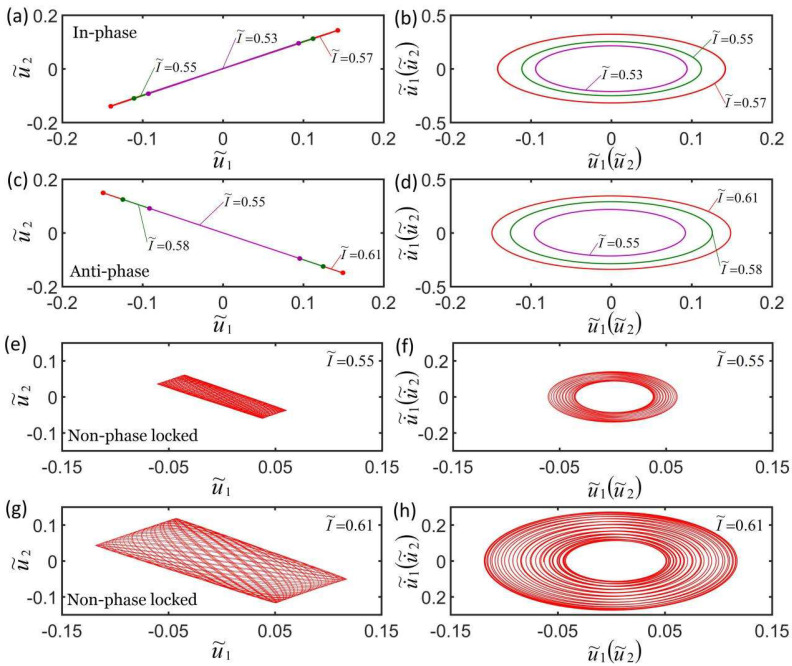
Effect of light intensity on self-oscillation for a constant synchronous mode. (**a**,**b**) Effect of light intensity on in-phase mode for ${\tilde{K}}_{f}=5$, $\tilde{c}=0.1$, $\lambda_{p}=1.15$, ${\tilde{K}}_{s}=0.3$, $C_{0}=0.7$, ${\tilde{u}}_{1}^{0}=0$, ${\tilde{u}}_{2}^{0}=0$, ${\tilde{\dot{u}}}_{1}^{0}=-0.3$, and ${\tilde{\dot{u}}}_{1}^{0}=-0.2$. (**c**,**d**) Effect of light intensity on the anti-phase mode for ${\tilde{K}}_{f}=5$, $\tilde{c}=0.1$, $\lambda_{p}=1.15$, ${\tilde{K}}_{s}=0.1$, $C_{0}=0.7$, ${\tilde{u}}_{1}^{0}=0$, ${\tilde{u}}_{2}^{0}=0$, ${\tilde{\dot{u}}}_{1}^{0}=-0.3$, and ${\tilde{\dot{u}}}_{1}^{0}=0.3$. (**e**–**h**) Effect of light intensity on the non-phase-locked mode for ${\tilde{K}}_{f}=5$, $\tilde{c}=0.1$, $\lambda_{p}=1.15$, ${\tilde{K}}_{s}=0.2$, $C_{0}=0.6$, ${\tilde{u}}_{1}^{0}=0$, ${\tilde{u}}_{2}^{0}=0$, ${\tilde{\dot{u}}}_{1}^{0}=-0.3$, and ${\tilde{\dot{u}}}_{1}^{0}=0.1$. Light intensity only affects the amplitudes of the three synchronization modes.

**Figure 7 micromachines-13-00271-f007:**
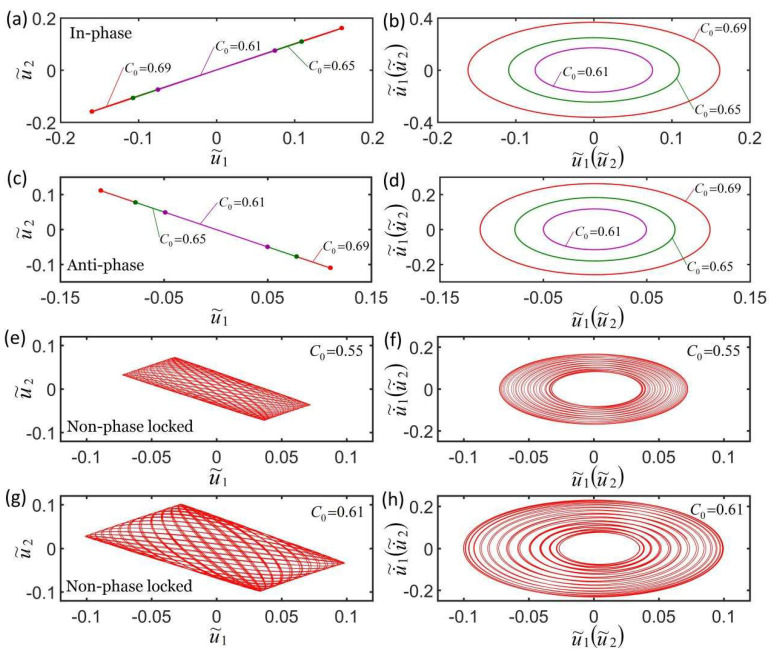
Effect of contraction coefficient on the self-oscillation for a constant synchronous mode. (**a**,**b**) Effect of three different contraction coefficients on self-oscillation in in-phase mode. The parameters are ${\tilde{K}}_{f}=5$, $\tilde{c}=0.1$, $\lambda_{p}=1.15$, ${\tilde{K}}_{s}=0.2$, $\tilde{I}=0.6$, ${\tilde{u}}_{1}^{0}=0$, ${\tilde{u}}_{2}^{0}=0$, ${\tilde{\dot{u}}}_{1}^{0}=-0.3$, and ${\tilde{\dot{u}}}_{1}^{0}=-0.2$. (**c**,**d**) Effect of three different contraction coefficients on self-oscillation in anti-phase mode. The parameters are ${\tilde{K}}_{f}=5$, $\tilde{c}=0.1$, $\lambda_{p}=1.15$, ${\tilde{K}}_{s}=0.2$, $\tilde{I}=0.6$, ${\tilde{u}}_{1}^{0}=0$, ${\tilde{u}}_{2}^{0}=0$, ${\tilde{\dot{u}}}_{1}^{0}=-0.3$, and ${\tilde{\dot{u}}}_{1}^{0}=0.3$. (**e**–**h**) Effect of the contraction coefficient on the self-oscillation in the non-phase-locked mode. The parameters are ${\tilde{K}}_{f}=5$, $\tilde{c}=0.1$, $\lambda_{p}=1.15$, ${\tilde{K}}_{s}=0.2$, $\tilde{I}=0.6$, ${\tilde{u}}_{1}^{0}=0$, ${\tilde{u}}_{2}^{0}=0$, ${\tilde{\dot{u}}}_{1}^{0}=-0.3$, and ${\tilde{\dot{u}}}_{1}^{0}=0.1$. The contraction coefficient only affects the amplitude of the three synchronization modes.

**Figure 8 micromachines-13-00271-f008:**
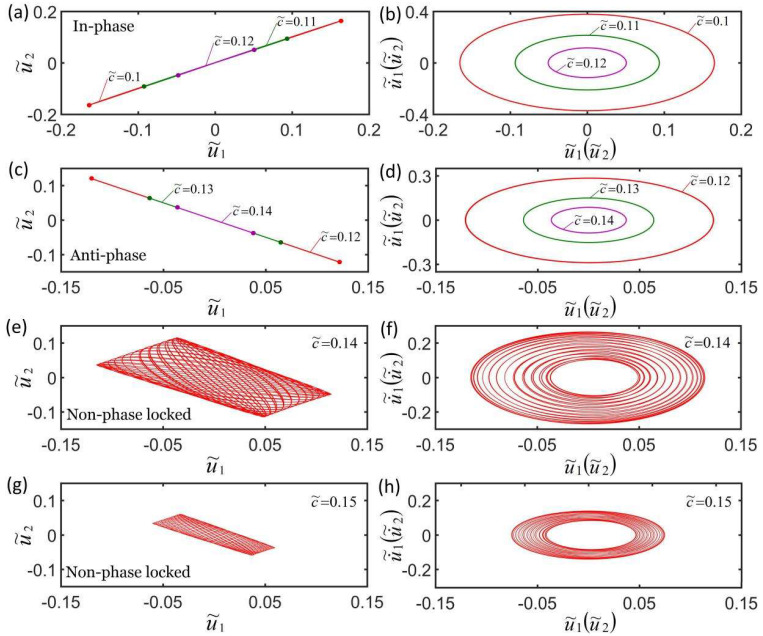
Effect of the damping coefficient on self-oscillation for a constant synchronous mode. (**a**,**b**) Effect of three different damping coefficients on self-oscillation in in-phase mode. The parameters are ${\tilde{K}}_{f}=5$, $\lambda_{p}=1.15$, ${\tilde{K}}_{s}=0.2$, $\tilde{I}=0.6$, $C_{0}=0.7$, ${\tilde{u}}_{1}^{0}=0$, ${\tilde{u}}_{2}^{0}=0$, ${\tilde{\dot{u}}}_{1}^{0}=-0.3$, and ${\tilde{\dot{u}}}_{1}^{0}=-0.2$. (**c**,**d**) Effect of three different damping coefficients on the self-oscillation in anti-phase mode. The parameters are ${\tilde{K}}_{f}=5$, $\lambda_{p}=1.15$, ${\tilde{K}}_{s}=0.2$, $\tilde{I}=0.8$, $C_{0}=0.7$, ${\tilde{u}}_{1}^{0}=0$, ${\tilde{u}}_{2}^{0}=0$, ${\tilde{\dot{u}}}_{1}^{0}=-0.3$, and ${\tilde{\dot{u}}}_{1}^{0}=0.3$. (**e**–**h**) Effect of the damping coefficient on the self-oscillation in the non-phase-locked mode for ${\tilde{K}}_{f}=5$, $\lambda_{p}=1.15$, ${\tilde{K}}_{s}=0.2$, $\tilde{I}=0.8$, $C_{0}=0.7$, ${\tilde{u}}_{1}^{0}=0$, ${\tilde{u}}_{2}^{0}=0$, ${\tilde{\dot{u}}}_{1}^{0}=-0.3$, and ${\tilde{\dot{u}}}_{1}^{0}=0.1$. The damping coefficient only affects the amplitude of the three synchronization modes.

**Figure 9 micromachines-13-00271-f009:**
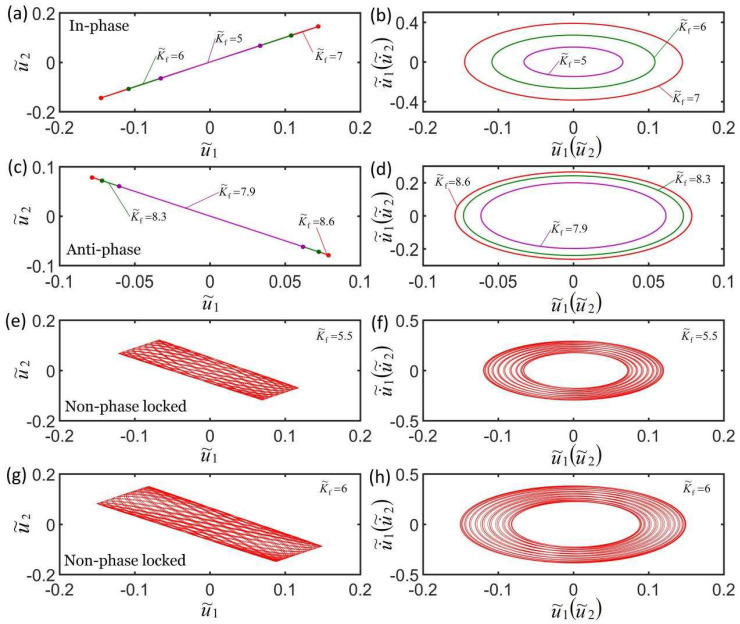
Effect of spring constant of LCE fiber on self-oscillation for a constant synchronous mode. (**a**,**b**) Effect of three different spring constants on self-oscillation in in-phase mode. The parameters are $\tilde{c}=0.1$, $\lambda_{p}=1.15$, ${\tilde{K}}_{s}=0.3$, $\tilde{I}=0.5$, $C_{0}=0.7$, ${\tilde{u}}_{1}^{0}=0$, ${\tilde{u}}_{2}^{0}=0$, ${\tilde{\dot{u}}}_{1}^{0}=-0.3$, and ${\tilde{\dot{u}}}_{1}^{0}=-0.1$. (**c**,**d**) Effect of three different spring constants on self-oscillation in anti-phase mode. The parameters are $\tilde{c}=0.1$, $\lambda_{p}=1.15$, ${\tilde{K}}_{s}=0.3$, $\tilde{I}=0.6$, $C_{0}=0.7$, ${\tilde{u}}_{1}^{0}=0$, ${\tilde{u}}_{2}^{0}=0$, ${\tilde{\dot{u}}}_{1}^{0}=-0.3$, and ${\tilde{\dot{u}}}_{1}^{0}=0.3$. (**e**–**h**) Effect of spring constant of the LCE fiber on the self-oscillation in non-phase-locked mode for $\tilde{c}=0.1$, $\lambda_{p}=1.15$, ${\tilde{K}}_{s}=0.3$, $\tilde{I}=0.5$, $C_{0}=0.7$, ${\tilde{u}}_{1}^{0}=0$, ${\tilde{u}}_{2}^{0}=0$, ${\tilde{\dot{u}}}_{1}^{0}=-0.3$, and ${\tilde{\dot{u}}}_{1}^{0}=0.2$. The spring constants of the LCE fiber have effects on the frequency and amplitude of the three synchronization modes.

**Figure 10 micromachines-13-00271-f010:**
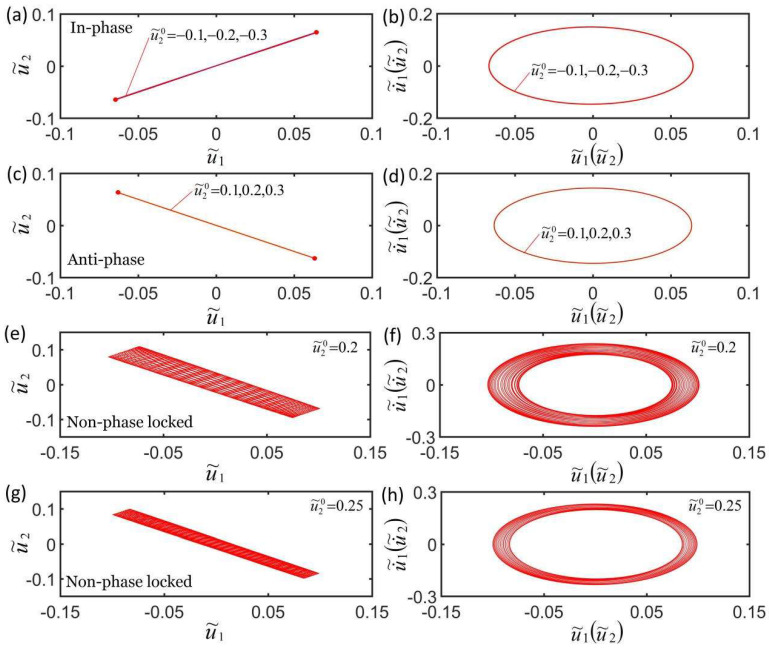
Effect of the initial condition on self-oscillation for a constant synchronous mode. (**a**,**b**) Effect of initial condition on self-oscillation in in-phase mode for ${\tilde{K}}_{f}=5$, $\tilde{c}=0.1$, $\lambda_{p}=1.15$, ${\tilde{K}}_{s}=0.2$, $\tilde{I}=0.5$, $C_{0}=0.7$, ${\tilde{u}}_{1}^{0}=0$, ${\tilde{u}}_{2}^{0}=0$, and ${\tilde{\dot{u}}}_{1}^{0}=-0.3$. (**c**,**d**) Effect of initial condition on the self-oscillation in anti-phase mode for ${\tilde{K}}_{f}=5$, $\tilde{c}=0.1$, $\lambda_{p}=1.15$, ${\tilde{K}}_{s}=0.05$, $\tilde{I}=0.5$, $C_{0}=0.7$, ${\tilde{u}}_{1}^{0}=0$, ${\tilde{u}}_{2}^{0}=0$, and ${\tilde{\dot{u}}}_{1}^{0}=-0.3$. (**e**–**h**) Effect of initial condition on the self-oscillation in non-phase-locked mode for ${\tilde{K}}_{f}=5$, $\tilde{c}=0.1$, $\lambda_{p}=1.15$, ${\tilde{K}}_{s}=0.2$, $\tilde{I}=0.5$, $C_{0}=0.7$, ${\tilde{u}}_{1}^{0}=0$, ${\tilde{u}}_{2}^{0}=0$, and ${\tilde{\dot{u}}}_{1}^{0}=-0.3$. The initial conditions affect the amplitude of the non-phase-locked mode and the change of the amplitude within a doubling period.

**Figure 11 micromachines-13-00271-f011:**
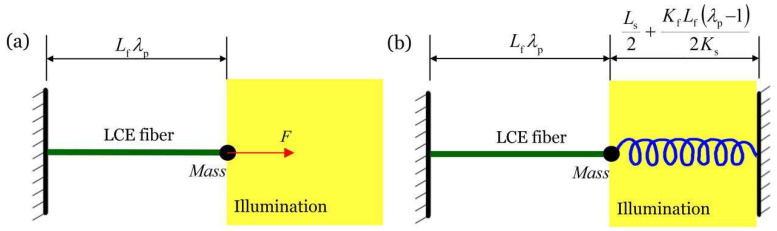
Equivalent systems of (**a**) in-phase mode and (**b**) anti-phase mode. In in-phase mode, the system is equivalent to a single oscillator with a constant force. In anti-phase mode, the system is equivalent to a single oscillator constrained by a fixed spring with half original length.

**Table 1 micromachines-13-00271-t001:** Material properties and geometric parameters.

Parameter	Definition	Value	Units
$L_{f}$	Original length of the LCE fiber	0.1	m
$K_{f}$	Internal radius	20~50	N/m
$K_{s}$	Coupling stiffness	0~5	N/m
m	Mass	0.5	g
c	Damping coefficient	5~8×10−3	kg/s
T0	Thermal relaxation time	10−2	s
η0	Light-absorption constant	3×10−3	1/s
I	Light intensity	1~3	W/cm2
C0	Contraction coefficient	0.4~0.7

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
