# Peer review of "Self-Sustained Collective Motion of Two Joint Liquid Crystal Elastomer Spring Oscillator Powered by Steady Illumination"

_micromachines, 2022, doi:10.3390/mi13020271_

Round 1

Reviewer 1 Report

This manuscript describes oscillators based on spring-connected LCE stripes. Various influential factors on the oscillation behaviors of the LCE-spring combined system were investigated. The conclusion was strongly supported by analysis. I think this work is publishable after some minor revisions. 

(1) Some English errors should be corrected, including spelling errors and incorrect tense.

(2) In the Introduction part, some important literatures about the LCE oscillators and LCE actuators were missed. Like Adv. Mater. 2020, 32, 1906319;  Adv. Funct. Mater. 2020, 30, 2000252; 

Reviewer 2 Report

Yu et al. report a numerical calculation of liquid crystal elastomer system showing self-sustained collective oscillation under steady illumination. The authors demonstrate that the system enables to exhibit of three steady synchronization modes of in-phase, anti-phase, and non-phase locked mode, and these phases can be controlled merely by changing some physical parameters of the system used. Such a self-oscillation system is abundant in nature and very important for realizing next-generation applications in soft-robotics fields. However, some experimental conditions are not reasonable and perhaps problematic in the LCE system. The following points should be clear.

  1. In line 132, the authors assume that the spring forces of LCE fibers are homogeneous and constant in the fiber and integrate equation 2, but they illuminated the fiber by using spatially structured light, meaning that the contraction of LCE fiber should be localized. Why can they assume the LCE fiber are homogeneous?
  2. The numerous calculation results are interesting, but I am not quite sure what kind of materials system they have assumed. This point might be important because some physical parameters are drastically affected by the system. In the manuscript, they employed the physical parameters of the system in lines 203–206 on Page 7 by citing the article of “Marshall, J.E.; Terentjev, E.M. Photo-sensitivity of dye-doped liquid crystal elastomers. Soft Matter 2013, 9, 8547- 639 8551”. However, some parameters they employed are different from the article, or also different from typical LCE systems as listed below.
  3. The value of Kf is 500–800 N m−1, but in my knowledge typical LCE shows ~ 100 N m−1.
  4. The value of I0 is 104–105 W m−2 (10–100 W cm−2), but light-responsive azobenzene-based LCEs can cause photo-isomerization by using ~10 mW cm−2

So, please mention what kind of LCE system they assume.

  1. They investigated the effect of physical parameters on oscillation modes. A slight change of the parameter allows one to tune the oscillation modes, but it is a little bit difficult to understand what change affects the modes. This means that the parameters are shown in sentences, but I recommend showing them in the table for ease of comparison.

Round 2

Reviewer 2 Report

The authors have done a nice job of addressing this reviewer's comments. The work is clear.